# Structural and Functional Annotation of Transposable Elements Revealed a Potential Regulation of Genes Involved in Rubber Biosynthesis by TE-Derived siRNA Interference in *Hevea brasiliensis*

**DOI:** 10.3390/ijms21124220

**Published:** 2020-06-13

**Authors:** Shuangyang Wu, Romain Guyot, Stéphanie Bocs, Gaëtan Droc, Fetrina Oktavia, Songnian Hu, Chaorong Tang, Pascal Montoro, Julie Leclercq

**Affiliations:** 1CIRAD, UMR AGAP, F-34398 Montpellier, France; shuangyangwu@foxmail.com (S.W.); stephanie.sidibe-bocs@cirad.fr (S.B.); gaetan.droc@cirad.fr (G.D.); pascal.montoro@cirad.fr (P.M.); 2AGAP, University Montpellier, CIRAD, INRA, Institute Agro, F-34398 Montpellier, France; 3State Key Laboratory of Integrated Management of Pest Insects and Rodents, Institute of Zoology, Chinese Academy of Sciences, Beijing 100101, China; 4University of Chinese Academy of Sciences, Beijing 100049, China; husn@im.ac.cn; 5Institut de Recherche pour le Développement, University Montpellier, UMR DIADE, 34394 Montpellier, France; romain.guyot@ird.fr; 6Department of Electronics and Automatization, Universidad Autónoma de Manizales, 170001 Manizales, Colombia; 7South Green Bioinformatics Platform, Bioversity, CIRAD, INRA, IRD, F-34398 Montpellier, France; 8Indonesian Rubber Research Institute, Sembawa Research Centre, 30953 Palembang, Indonesia; fetrina_oktavia@yahoo.com; 9State Key Laboratory of Microbial Resources, Institute of Microbiology, Chinese Academy of Sciences, Beijing 100101, China; 10College of Tropical Crops, Hainan University, Haikou 570228, China; chaorongtang@hainanu.edu.cn

**Keywords:** transposable elements, siRNA, rubber tree, transcriptional regulation, epigenomics

## Abstract

The natural rubber biosynthetic pathway is well described in *Hevea*, although the final stages of rubber elongation are still poorly understood. Small Rubber Particle Proteins and Rubber Elongation Factors (SRPPs and REFs) are proteins with major function in rubber particle formation and stabilization. Their corresponding genes are clustered on a scaffold1222 of the reference genomic sequence of the *Hevea brasiliensis* genome. Apart from gene expression by transcriptomic analyses, to date, no deep analyses have been carried out for the genomic environment of *SRPPs* and *REFs* loci. By integrative analyses on transposable element annotation, small RNAs production and gene expression, we analysed their role in the control of the transcription of rubber biosynthetic genes. The first in-depth annotation of TEs (Transposable Elements) and their capacity to produce TE-derived siRNAs (small interfering RNAs) is presented, only possible in the *Hevea brasiliensis* clone PB 260 for which all data are available. We observed that 11% of genes are located near TEs and their presence may interfere in their transcription at both genetic and epigenetic level. We hypothesized that the genomic environment of rubber biosynthesis genes has been shaped by TE and TE-derived siRNAs with possible transcriptional interference on their gene expression. We discussed possible functionalization of TEs as enhancers and as donors of alternative transcription start sites in promoter sequences, possibly through the modelling of genetic and epigenetic landscapes.

## 1. Introduction

Natural rubber biosynthetic pathway was very recently reviewed by [1]. *Hevea brasiliensis* is a sole tropical perennial crop used for the industrial production of natural rubber (NR) [2]. Selected clones, for latex and/or timber production, are propagated by grafting. The cis-1,4 polyisoprene is biosynthesized from sucrose produced by photosynthesis in the leaves and translocated to specialized cells called laticifers. After loading, sucrose is metabolized into isopentenyl pyrophosphate (IPP), a monomer used for elongation of the polymer biosynthesized in the rubber particles of latex cells [3]. All genes involved in the NR biosynthesis pathway have been identified in the genomic sequences of the Chinese rubber clone Reyan 7-33-97 [4], and particularly the genes encoding the Rubber Elongation Factor (*REF1–8*) and Small Rubber Particle Protein (*SRPP1–10*) families. *SRPPs* were identified in many plants (rubber and non-rubber producing), while *REFs* were not found in other plants [1]. They have been shown to be more extended in *Hevea* compared to other rubber-producing plants [4,5], probably due to a whole genome duplication shared with cassava (*Manihot esculenta*), another *Euphorbiaceae* plant species [6]. Among various clones, *REF* transcript levels have been shown to be positively correlated with latex yield [7]. In addition, clonal phenotypic properties exit as they are classified according to the biochemical ability of latex cells to produce rubber assessed by their sucrose, thiols and inorganic phosphorus contents. Sucrose content reflects the balance between sucrose consumption by the latex cells for energy production, latex biosynthesis, and the transfer of sucrose from the apoplast to the latex cells. Inorganic phosphorus content indicates the intensity of metabolic activity in the latex cells [8,9]. The thiol content reflects the redox status of the tree, namely the level of control of the oxidative stress of trees under production [10,11,12,13]. These three parameters are actually used for a good management of the plantation such as tapping frequency and ethephon stimulation [14]. In case of overexploitation of *Hevea* by excessive tapping or stimulation with ethephon, physiological disorders appear leading to Tapping Panel Dryness (TPD). This clone-dependent physiological disease [15,16] is characterized by oxidative stress in laticifers, leading to a cessation of latex flow with in situ coagulation of rubber (for review [13,17]).

In addition to biochemical data, we have access to a transcriptomic data. For example, the transcripts *REF1, REF3, REF7* and *SRPP1* are strongly expressed in the laticifers in the clone Reyan 7-33-97. In contrast, transcripts from other families of *SRPPs* (*SRPP3, SRPP5, SRPP8* and *SRPP9*) and *REFs (REF2, REF4, REF5, REF6* and *REF8*) were detected at low levels in latex. By contrast, in clone RRIM 600, *REF2, REF3, REF7, REF8* and *SRPP1* are the most highly expressed in latex [18]. Beyond a differential contribution of *REF* and *SRPP* isoforms to NR biosynthesis between clones, possible transcriptional control has been suggested, as in clone RRIM 600 some of the natural rubber biosynthetic genes displayed an alternative transcription start site (TSS), in a tissue-dependent manner [18].

Several *de novo* genomes of rubber clones, with contrasting latex physiology, are available such as Reyan 7-33-97 [4], RRIM 600 [18], BPM 24 [19]; PB 260 [20,21] and GT 1 [5]. The rubber tree genome is large and complex to assemble because of its high heterozygosity, whole genome duplication and transposable element (TE) content [4,5]. TEs, first described in 1931 by the Nobel Prize winner Barbara McClintock, are also known as “jumping genes”. Transposition is often neutral, but their mutagenic potential is now demonstrated through genetic and epigenetic mechanisms, in the establishment and/or alteration of gene regulatory networks, in the modification of phenotypes and in the possible generation of adaptive genetic variations [22]. TE exaptation, which is the transformation of the repeated elements into a novel host gene, or a promoter region, is consistent with their role as “engines of plant genome evolution” [23,24]. TEs are classified hierarchically by subdividing them into Classes, Orders, Super-families, Lineages and Families [25]. Class I, corresponds to retrotransposons (LTR (Long Terminal Repeat), PLE, LINE (Long Interspersed Nuclear Element) and SINE (Short Interspersed Nuclear Element)), while Class II comprises DNA transposons (TIR (Terminal Inverted Repeat), Crypton, Helitron and Maverik). This classification system has been successfully used in the TE classification of several plant species [25,26,27,28,29,30,31]. In plant genomes, LTR retrotransposons are present in large numbers and classified according to the protein sequence [32], with two evolutionary distinct Superfamilies, *Ty3/gypsy* (*Athila, Tat, Galadriel, Reina, CRM/CR*, and *Del/Tekay* among others) and *Ty1/copia* (*TAR/Tork, Angela/Tork, Ikeros/Tork, Maximus/Sire, Ivana/Oryco, Ale/Retrofit* and *Bianca*, among others) [33]. However, the transcription and movement of these retroelements have been observed in response to development and environmental cues, due to the presence of *cis*-regulatory patterns, which could facilitate plant adaptation by giving new regulatory patterns to genes [23]. Small interfering RNAs (siRNAs), which trigger RNA-directed DNA methylation (RdDM), effectively silence TEs [34]. Our previous studies on populations of small RNAs in *Hevea* clone PB 260 showed alterations in the microtranscriptome of laticifers in response to environmental cues and TPD [35]. A shift in the distribution of small RNAs has been observed, with peaks of small RNAs of 24- and 21-nt, in healthy and TPD-affected trees of clone PB 260, respectively. This suggests a shift from transcriptional regulation in healthy trees to post-transcriptional regulation in trees affected by TPD [35,36,37]. Very recently, it has been shown that 21-nt small RNAs do not originate from *MIR* genes and could therefore be classified as siRNAs [20]. In addition, given the size of these siRNAs (21-nt), they could be responsible for the degradation of target transcripts by post-transcriptional cleavage, or the maintenance of TE silencing by epigenetically activated small interfering RNAs (easiRNAs) [34].

In this study, we chose PB 260 clone for which genomic [20,21], transcriptomic [17,38,39] and microtranscriptomic data [20,35,37] were available (Figure 1) [4,5,17,18,20,21,35,37,38,39]. In addition, functional validation could be possible for this clone as the genetic transformation procedure is robustly mastered [40,41,42,43,44,45]. Our objective was to annotate, in clone PB 260, TEs and siRNAs sharing sequence identity with TEs (namely TE-derived siRNAs as a track of annotation), to infer their possible interference with gene expression in laticifers, and to show TE-transcriptional regulation of genes involved in rubber biosynthesis. Firstly, we carried out in silico detection and accurate annotation on a genome scale of TEs with the REPET and LTR_STRUC pipelines [46] (Figure 2), as only a few active elements have been described in the genus *Hevea* so far [47]. Thus, from the *Hevea* TE database generated, their diversity and activity were further analysed (Figure 1). Secondly, as the majority of the *REF/SRPP* genes were clustered on a single scaffold 1222 [4], this genomic region was curated manually. The capacity of TEs to produce siRNAs was evaluated. Lastly, all this computational information was combined to put forward hypotheses on how genomic/epigenomic environments affect the transcriptional regulation of *REF/SRPP* gene cluster by comparing side-by-side genomic sequences from other clones (PB 260, Reyan 7-33-97, RRIM 600 and GT 1).

## 2. Results

### 2.1. Transposable Element Annotation Process

To accurately annotate TEs in the re-sequenced genome of rubber clone PB 260, the LTR_STRUCT and REPET pipelines were used (Figure 2). LTR_STRUCT detected 3637 sequences from the 84,128 blocks in the re-sequencing data. Fifty-eight LTR consensus sequences were built and the singletons (1785) were stored in the expert bank. At the same time, 300 Mb of genomic sequences were analysed by the REPET software, which includes the TEdenovo and TEannot pipelines (Figure 2).

TEdenovo detected 7655 consensus sequences, which were added to the expert bank already containing the consensuses obtained by LTR_STRUCT (9448 sequences in total). Three TEannot rounds were required. The first two rounds were used to construct libraries of full-length fragments of transposable elements (4385 sequences) and full-length copies of transposable elements (4388 sequences), finally resulting in the detection of 2189 consensus sequences after the second round and the manual curation steps. The third round of analyses was performed to annotate the entire genome, leading to the detection of 954,090 TEs, representing 75% of the genome. After filtering TEs at least 80 bp long and sharing a homology of at least 80% [25], 448,210 TEs were fully annotated, representing 56% of the genome (Figure 2 and Table 1).

### 2.2. Composition of the Genome in TEs, Their Diversity and Activity

Complete and incomplete TEs were classified according to their Orders and Superfamilies, as described in [25]. In rubber, the largest Order consists of the LTR retrotransposons (RLX in Table 1). LTRs represent more than 77% of TE length in the genome sequence of clone PB 260. Far behind, the Large Retro-transposon Derivatives (RXX-LARD) Order/Superfamily represented 7%. MITE (Miniature Inverted-repeat Transposable Element, 0.13% of TE) and SINE (0.17% of TE), with less than 1700 copies each, were annotated separately with more specific pipelines, namely MUST and Sine_finder (Figure 2). Since most TEs belonged to the LTR Order, a complete phylogenetic analysis, based on amino acid sequences from the reverse transcriptase (RT) domain, when present, was carried out on 1476 full-length copies obtained after LTR_STRUC and Inpactor classification with nine major lineages of LTR retrotransposons, of which 4 and 5 belonged to the *Ty1/copia* and *Ty3/gypsy* superfamilies, respectively (Appendix A). In addition, the representativeness of the diversity of LTRs was not affected by the manual curation step of the consensus sequences predicted with REPET, as shown by the complete phylogenetic analysis carried out before and after manual curation (Appendix A). Lastly, at genome-wide level, a significant fraction of the *Hevea* genome was composed of LTR *Ty3/gypsy* elements and more particularly the *Del/Tekay* groups, with more than 9129 elements (9129/11,989; 76.1% of RT domains) (Figure 3).

LTR transposition is based on the production of polyadenylated RNA molecules through the presence of a site for *RNA pol II* inserted in the 5′ UTR region and a polyadenylation site in the 3’ UTR region [48]. The timing of insertion of full-length LTR retrotransposons was measured by calculating the divergence of LTRs of the same elements and estimating the time using a substitution rate for *Hevea*. The most recently inserted elements shared a high level of similarity between their LTRs. Older inserted elements showed more mutations in their sequences. Each *superfamily* and lineage of full-length retroelements was characterized and their insertion times estimated (Figure 4). As shown in Figure 4A, the *Ty3/gypsy* and *Ty1/copia* superfamilies had different insertion times, *Ty1/copia* showing more recent insertions than *Ty3/gypsy*. At lineage level, no *Ty3/gypsy* lines had recently been inserted despite the massive presence of LTRs from the *Del/Tekay* line (Figure 4B), while the *Retrofit/Ale* lines were the most recently added *Ty1/copia* elements (Figure 4C). In the available RNA-seq data from latex [39], none was found to be differentially expressed in response to ethephon or ethephon-induced TPD (Appendix A).

### 2.3. Annotation of TE-Derived siRNAs

Small RNAs aligning with TE sequences were further annotated as TE-derived siRNAs then classified according to their size (Figure 5, Appendix A). The majority of siRNAs derived from TEs had a size of 23 and 24 nt (above 35% of total reads), followed by size classes of 22 (15%) and 21 nt (10%). Conversely to previous assumptions, no differences in the distribution of siRNAs derived from TEs were observed between young seedlings, healthy trees and trees affected by TPD (Figure 5).

The production of 24-nt siRNA by TE superfamilies was then quantified by counting those that mapped on TE sequences (Figure 6, Appendix A). The LTR retrotransposon order produced the largest number of siRNAs (70%), with 33% of them by *Ty3/gypsy* -RLG, followed by unclassified retrotransposon RIX (18%) and *Ty1/copia* RLC (16%). The DTX of DNA transposons contributed to about 3% of siRNA production. We further analysed the 24-nt siRNAs production by TEs in three small RNA-seq libraries including young plants, latex from healthy trees and latex from TPD-affected trees [35,37]. Interestingly, there was no significant in siRNAs production by transposable elements between young trees, latex from healthy and TPD-affected trees. The same proportions were observed for 21, 22 and 23 nt (Appendix A).

To look into possible implication of in post-transcriptional regulation, degradome data were analysed with TE-derived siRNA data sets (20–22 nt), excluding previously identified miRNAs [20]. None of the TE-derived siRNAs with a size of 20–22 nt demonstrated post-transcriptional activity as well as for 24-nt TE-derived siRNA as expected (data not shown).

### 2.4. Structural and Functional Annotation of a Re-Sequenced Genome from Clone PB 260

To transfer the structural and functional annotation from the Reyan 7-33-97 reference genome to clone PB 260, the EGN-EP transfer pipeline was used [49]. The transfer was successful for 40,241 of the 46,710 Reyan 7-33-97 genes, concerning 3452 scaffolds. The BUSCO evaluation of the completeness of genome annotation showed that EGN transfer was not effective enough to annotate the re-sequenced PB 260 genome with only 81.6% of the complete genes (Appendix A).

The EGN-EP pipeline was therefore used to predict a full genome annotation (Appendix A). In this case, 164,180 genes were detected, with a BUSCO score of 91% complete genes (Appendix A). The large number of predicted genes was probably due to gene fragmentation related to assembly errors, or to transposable element genes (the capsid Gag protein and the polyprotein Aspartic proteinase-RT-RNAseH). A rubber tree genome with such a high content of repetitive elements can generate assembly errors, as well as high heterozygosity, which does not allow for the differentiation of alleles and gene duplications.

Thus, after post-filtering on doubtful short genes and transposable element genes, 57,181 genes were kept with a BUSCO score of 90.2% complete genes (Appendix A).

### 2.5. Transcriptional Interference by TEs and siRNAs on Genes Involved in NR Biosynthesis

Based on the chromosomal coordinates of genes and transposable elements, their co-localization was searched. Eleven percent of genes harboured a TE at a ±1 kbp distance and 8.2% at ±500 bp. This proximity may interfere with the transcriptional regulation of genes. To study this interference on genes involved on NR biosynthesis, we focused on scaffold 1222 (205 kbp), containing most of the genes involved in the biosynthesis of natural rubber [4].

After transfer of structural and functional annotations of the genomic sequences of clone PB 260, and after manual curation of scaffold 1222, a phylogenetic analysis was carried out with the full-length nucleotide sequences of *REF/SRPPs*. It revealed local duplication of the *SRPP9* and *REF8* genes (*SRPP9b*: Scaffold1222_36463_37765; *REF8b*: scaffold 1222_168119_168886) and a partial gene at the distal part of scaffold 1222 (scaffold1222_203928_2041979) (Figure 7).

After sequence comparison with the Reyan 7-33-97 genome, local duplications and the partial gene were also present but not reported (Figure 8). By analysing the primary sequence, all *REF/SRPP* genes were surrounded by TEs, either in their promoter, or in an intron, or after the 3’ UTR (Figure 8). According to Lau et al., 2016, *SRPP1, SRPP7, REF3, REF6, REF7, REF8* and *REF9* are associated with scaffold 1741 in clone RRIM 600. PB 260 and RRIM 600 scaffold sequences were compared side by side (Figure 8). The region containing *SRPP1, SRPP3, SRPP8, SRPP9a, SRPP9b, REF1, REF2* and *REF7* and their promoters were retained in both clones. Concerning gene expression, in RRIM 600, the highest *REF* transcript expressed was *REF3* and not *REF1* [18]. In the *REF3* genomic region, there was no conservation between RRIM 600 and PB 260 (Figure 8), which could explain the difference in expression level. Conversely, comparative analysis between scaffold 1222 from PB 260 and chromosome 9 from the GT 1 genome indicated a deep restructuring of the *REF/SRPP* locus in GT 1 (Figure 8). In GT 1, the *REF/SRPP* genes were distributed in two loci on chromosome 9 separated by 12 Mb (13.2–13.5 Mb and 1.5–1.6 Mb), which indicate the large difference in *REF/SRPP* genes between these two cultivars. These results may reflect the difference in the ability of two cultivars in rubber biosynthesis.

It is interesting to note that the *SRPP1* gene, strongly expressed in latex with more than 120,000 counts (Table 2), contained a LTR/*Ty1/copia/Retrofit/Ale* (TE6832), almost complete, for which the 3’ part of the polyprotein and the 3’ LTR were missing in its promoter, without siRNA production. TE6832 was not differentially expressed in response to ethephon-induced TPD (Appendix A). On the other hand, when a large amount of siRNA was detected upstream or downstream of a gene body, the level of expression was considerably reduced, such as *SRPP9, SRPP9b* and *SRPP3,* with fewer than 110 counts in each of the biological repeats (Table 2).

For the *REF8b* gene, incomplete insertion of LTR was observed in the gene body, leading to the loss of the third exon. This gene is probably no longer functional, even if reads had been detected in the RNA-seq data (Figure 8, Table 2). *REF8b* had a much lower expression level than *REF8*, with fewer than 2500 and more than 4000 counts, respectively (Table 2). On the other hand, analyses of the RNA-seq data showed that the partial sequence, non-annotated and detected at the 3′ extremity of the scaffold, was not expressed at all (no count, Table 2).

*Cis*-regulatory elements in the promotor sequences were detected in highly, moderately and weakly expressed gene (Table 3). The strong expression of the *SRPP1* gene associated with the presence of a large number of enhancer pattern (14) brought by the transposon located in the promoter region. The *REF1* and *REF8* genes have an identical number of enhancer motifs (4) and an expression level varying by a factor of approximately 80, which is not explained by the presence of other motifs. We noted the presence of siRNAs mapping on the exons of the *REF1* gene, and to a much lesser extent for the exons of the *REF8* gene. It is worth mentioning that *cis*-regulatory elements of expression induction under anaerobic conditions were detected in all the promoters tested. However, a diversity of hormone response boxes was observed with members responding to auxin (*REF3–8* and *SRPP5*), ABA (*REF2–5* and *SRPP*1), MeJA (*REF*8) and salicylic acid (*REF3* and *REF4*) (Table 3).

## 3. Discussion

The purpose of this study was investigating the influence of transposons on the expression of genes located in their vicinity. We precisely annotated the transposable elements to identify all siRNAs sharing a sequence identity with them. Further analysis was performed on scaffold 1222 containing a cluster of genes involved in rubber biosynthesis.

### 3.1. Consequences of TE Location and siRNA Production Nearby in Genes Involved in Natural Rubber Biosynthesis

The scaffold 1222 sequence in clone PB 260 was constructed from clone Reyan 7-33-97 de novo genome. In clone RRIM 600, local variations in gene location were observed in the corresponding scaffold (scaffold 1741 [18] corresponding to contig MKXE01001735.1 in the repository). Conversely, the GT 1 genome showed deep restructuring of the *REF/SRPP* locus.

The systematic presence of TEs in the promoter sequences observed on scaffold 1222 questioned the regulation of *REF* and *SRPP* gene expression. The presence of TEs in the promoter region may confer regulatory *cis*-elements, functionalizing the TEs as part of the promoters, which could also explain variable expression in a tissue-dependent manner. In addition, TEs may also bring alternative transcription start sites (TSS), as shown for clone RRIM 600, where 1/3 of alternative TSS, identified by 5′-CAP sequencing (CAGE), were located no more than two kbp upstream of the transcript at genome level [18]. Alternative TSS may therefore allow the generation of variable protein isoforms for the N-terminal region and long non-coding RNAs [50]. An epigenetic mechanism is at play to repress alternative TSS, based on the methylation of chromatin changes, which is required to maintain precise gene expression control, as demonstrated in eukaryotic genomes [51]. Chromatin marks have the potential to redefine the promoter as an intragenic region, allowing alternative TSS to generate variable protein isoforms for the N-terminal region and/or long non-coding RNAs [50]. In rubber clone PB 260, the next step will be to functionally validate the link between chromatin marks, the presence of TEs, and siRNA production in the environment of genes to demonstrate the transcriptional control of gene expression. Given the large number of TEs in the genome, environmental, developmental and clonal discrepancies in alternative TSS are expected when looking for transcript overlapping promoters and annotating long non-coding RNAs.

TEs are maintained in silent form either by changes in the chromatin structure or directly by cytosine methylation of their sequences in CG, CHG and CHH contexts [52]. DNA methylation in a CHH context is the signature of RNA-directed DNA methylation, RdDM [53,54]. Variation in patterns of methylation within genes and surrounding sequences are associated with a continuous range of expression differences, from silencing to constitutive expression [55]. Methylation around TSS is associated with the silencing of expression [55]. In tomato, a peak of 24-nt siRNA -500 kb upstream of the TSS distribution induced high mCHH [56]. In the rubber, we showed a negative link between siRNA production in the gene body and transcriptional activity of the corresponding gene. It can suggest that the large variation in the level of expression between *REF1*, highly expressed in latex, and *SRPP9, SRPP9b* and *SRPP3*, poorly expressed in latex and associated with siRNA production in promoter regions, may be related to a level of CHH methylation. Methylation levels will be assessed on the genomic DNA of bark, comprising latex cells. The collection of latex, corresponding to the cytosol of this specialized cell, only provides easy access to ribonucleic acids (RNAs and small RNAs), as the nuclei remain attached inside.

### 3.2. TE Diversity in Hevea Clones

All available genomic sequences are from the unrelated Wickam genetic group, with the exception of BPM 24, which is from a GT 1 cross and Reyan 7-33-97 from a RRIM 600 one (http://rubberclones.cirad.fr/). In the de novo genomes of rubber clones Reyan 7-33-97 [4], RRIM 600 [18], BPM 24 [6] and GT 1 [5], the repeated regions represent 66%, 69%, 71% and 70% of the genome, respectively, of which 64% to 67% comprise LTR retrotransposons. In contrast, the ratio between the two superfamilies *gypsy* and *Copia* greatly differed between clones with 6.05 for Reyan 7-33-97, 3.71 for RRIM 600, 3.69 for clone BPM 24 and 3.12 for GT 1. In the present study, the content of repeated elements was higher (75%) in clone PB 260, of which LTR retrotransposons represented 43% of annotated transposons. The *gypsy*/*Copia* ratio was slightly lower (2.24) in clone PB 260 than other rubber clones. The diversity of TE content and *gypsy*/*Copia* ratio between rubber clones may be due to the methods used to identify and classify repeated sequences. Indeed, RepeatMasker and RepeatModeler were used for the annotation of the genome from Reyan 7-33-97, BPM 24 and RRIM 600, while the REPET pipeline, combined with LTR_STRUCT, was used for clone PB 260. REPET pipeline was chosen for its sensitivity, by connecting TE fragment even from afar, and its specificity in the detection of TE [46].

The latest reference genome, clone GT 1, was annotated with LTR_FINDER combined with RepeatMasker [5]. As for clone GT 1, the genome size of clone PB 260 increased with the massive expansion of the *Ty3/gypsy Del/Tekay* retroelements.

The size of the genome could also influence the TE content. The genome of Reyan 7-33-97 had an estimated size of about 1.46 Gb, while the size expected for the other clones, measured by flow cytometry, was about 2.15 Gb. For clone PB 260, only re-sequencing data are available and scaffolding was done by mapping reads on the Reyan 7-33-97 reference genome. Analysis bias was therefore possible for PB 260. The genome size and TE content variations between clones may also indicate a dynamic genome with recent events, such as proliferation and/or deletion of repeated sequences. However, these variations cannot be attributed to recent insertional activities of identified LTR retrotransposons, as demonstrated by insertion times in PB 260 and GT 1 [5]. The amplification of *Del/Tekay*, the most redundant LTR retrotransposon lineage that contributed to the increase in the genome size of rubber clone PB 260, occurred between 2 to 10 My, with peaks between 6 to 8 My. This observation may suggest that other forces, such as unequal recombination, deletion, or duplication may be involved in *Hevea* genome reshaping. This point deserves to be further studied, since several agro-physiological traits, such as the natural rubber production potential, reactivity to stimulation by ethephon, the characteristics of the polyisoprene chain lengths, and the ability to adapt to abiotic and biotic stress are clone dependent.

### 3.3. Post-Transcriptional Activity of TE-Derived siRNAs

We had previously identified 445 transcripts degraded by miRNAs [20]. In this study, we did not identify any targets that were post-transcriptionally regulated by TE-derived siRNAs, consistent with the literature, revealing only translation inhibition and miRNA mimicry. To date, only two kinds of siRNA activities had already been studied. The first one is in Arabidopsis, where *Athila* LTR-derived small RNA (siRNA854, 22 nt) was shown to bind the 3′UTR of gene *UBP1b* with subsequent translational inhibition [57]. The same mechanism was shown in a Drosophila embryo, where TE-derived small RNAs (piwi RNAs) bind to 3′UTR of *nos* mRNA, with subsequent translational inhibition, giving a protein gradient across the embryo for the proper segment fate [57]. In the case of miRNA mimicry, a long non-coding RNA is expressed by a retrotransposon in rice root, which acts as a decoy mRNA and which traps miRNA171. The removal of post-transcriptional regulation by miRNA171 in roots allows the synthesis of the master regulators of root development, SCARECROW-LIKE transcription factors [58]. In rubber trees, these two types of regulation, translational repression and miRNA mimicry, deserve to be validated by monitoring protein formation. This cannot be done on a large scale, but could be carried out on only a few targets of interest to demonstrate the link between the production dynamics of siRNAs produced by transposons, the identification of their corresponding targets for which proteins will no longer be formed, and the impact of the activation or suppression of post-transcription regulation on cell function in a given physiological context or developmental stage.

## 4. Materials and Methods

### 4.1. Clone PB 260 Nuclear Genome Re-Sequencing and Other Genomes Available Used in this Study

Nuclear DNA from leaves of rubber clone PB 260 was sequenced by GATC (https://www.eurofinsgenomics.eu/) using Illumina pair-end sequencing (2 × 150 bp) [21]. Briefly, a total of 84 Gb was obtained, from which 62 Gb were assembled through alignment against the clone Reyan 7-33-97 reference genome [4]. Unmapped sequences (~200 kbp) were further assembled using the MaSuRCA (Version 3.2.4) mega-reads algorithm [59], by using clone BPM 24 SMRT raw reads [6]. The PB 260 re-sequenced genome assembly used in this study was therefore a combination of two origins of mapped sequences. The re-sequencing data are available under project number PRJCA001333 in the GSA [36] and BIG Data Center [60]. The Overview of genome assembly available and used in this study is in Table 4.

### 4.2. Detection of TEs by TEdenovo and LTR_STRUC

From the re-sequenced PB 260 genome assembly, N stretches over 11 nt in length were removed by the dbChunk.py program installed in the REPET pipeline (Version 2.1) [46]. Genomic sequences from clone PB 260 were chunked into 84,218 contigs. Then, 300 Mb, representing the largest 4000 contigs, were used as inputs to the TEdenovo pipeline [61], as recommended to avoid the use of to many informatic resources for very repetitive sequences (V. Jamilloux, personal communication, [62]). TEdenovo consensus nucleotide sequences were classified according to the Repbase database [63] and named by the classification proposed by Wicker’s hierarchical TE classification system [64]. The PASTEC program [64] removed chimeric sequences (chim) and no category (nocat). An additional filter was added to delete sequences with a confidence interval (CI) <20. CdHit [65] was used to cluster sequences. Rpt_map in REPET was used to build alignments with a minimum of three sequences in one cluster and dbConsensus.py to obtain consensus in each cluster. Then, all the genomic sequences (1.2 Gb) were used to launch LTR_STRUC [66] with default parameters, in order to detect full-length LTR retrotransposons. Full-length LTR retrotransposon annotation and analyses were performed by INPACTOR [67]. Sequences without clustering were defined as singleton sequences. Combined consensus sequences, without redundancy, performed by the PASTEC program [64] from the three analyses (TEdenovo, LTR_STRUCT and singleton), were used as an input for TEannot.

### 4.3. TE Annotation

Three paths of the TEannot pipeline were needed (Figure 2). The first one, run on the same subset of 300 Mb of genomic sequences, allowed the annotation of full-length fragments and copies. Full-length fragments were used for a second path on the same subset of 300 Mb. A manual curation step was added before the third path. Briefly, alignments against the consensus sequences were checked individually to discard chimeric and badly covered consensuses, as well as low complexity sequences. To do so, a plot coverage and a dot-plot were generated for each consensus. Moreover, a phylogenetic analysis with the reverse transcriptase (RT) domains before and after manual curation allowed validation of the presence of one copy for each lineage and family. The third and last path was run on the 1.2 Gb genomic sequences from clone PB 260. Additional filters were added to delete sequences < 80bp with less than 80% homology [25]. PB260_TE.gff3 and PB260_TE_filtered.gff3 files were generated.

### 4.4. MITE and SINE Prediction

MITE detection was performed with the MUST program [68] and redundancy was removed by BLASTClust [69]. SINE detection was performed by SINE_ finder [70]. The PB 260 genomic sequences were annotated with Censor [71]. The corresponding SINE.gff3 and MITE.gff3 were generated.

### 4.5. Estimation of Insertion Time for LTR Retrotransposons

The insertion time of full-length copies was estimated based on the divergence of the 5′- and 3′-LTR sequences of each copy [72]. The difference in the Ka/Ks ratio between Cassava and *Hevea* is 0.14 (Cassava 0.37 and *Hevea* 0.23) [4] corresponding to a speciation time of around 36 million years ago dated thanks to a fossil-calibrated molecular clock for the *Euphorbiaceae* [6]. The insertion dates (T) were estimated using the formula *T*  =  Ks/2r, where T is the time of divergence, Ks is the number of synonymous base substitutions per site and r is the substitution rate [72]. The calculated average base substitution rates per year for coding sequences (r) in *Euphorbiaceae* could therefore be calculated as Ks/2T = [0.14/(2*36MY)] = 1.94 × 10^−9^. The rate used for the LTR divergence analysis was 2-fold higher (3.89 × 10^−9^) than that determined for coding sequences in *Hevea* based on the assumption that non-coding sequences evolve more rapidly [73,74,75,76]. Almost intact Target Site Duplication (TSD) at both ends of the elements were kept for the estimation of insertion time. Among them 79.4% show intact TSD, 19% contained one mismatch and 1.5% contained two mismatches.

### 4.6. TE-Derived siRNA Annotation and Abundance

Five small RNA-seq datasets are available for clone PB 260, from juvenile plants in the greenhouse (leaf, stem, root) to mature plants in the field (from young and mature leaves, from latex of healthy and TPD-affected trees) [35,37,77,78]. Cutadapt was used to remove adapter and low-quality reads. MITP (https://sourceforge.net/projects/mitp/) was used to identify the miRNAs [20,21]. The clean siRNA sequences were obtained by removing all predicted miRNAs in clean reads. Blat [79] was used to get the small RNA loci in the PB 260 genome with the parameter “minScore = 10, tileSize = 8”, and “blat_top_hit_extractor.pl” in Trinity [80] was used to ensure the best hit of the blat result. The BEDTOOLS program (2.24.0) was used to intersect the siRNA loci with that of the transposable elements. The PB260_TE_derivedsiRNAs.gff3 file was generated.

### 4.7. Gene Annotation of the Clone PB 260 Genome

EGN-EP transfer (egn_annotation_transfer.pl) [49] was used to transfer annotations from the reference genome from clone Reyan 7-33-97 to clone PB 260. The PB260_Gene.gff3 file was generated. Full EGN-EP was used to predict genes, combining the mapping of expression data (PB 260 transcriptomes db and NCBI *Euphorbiaceae* EST db) and polypeptide data (Swiss-prot db, Reyan 7-33-97_protein.faa [4] and TrEMBL db). The PB260_Gene_EGNP.gff3 file was generated. All annotation results were then evaluated by a BUSCO analysis [81].

Post-processing steps were necessary, on the one hand to filter out short (less than 600 bp) and doubtful genes, having either no BLAST hit with the protein databanks (/status = obsolete) or a BLAST hit in Repbase (/transposable_element_gene = 1), and on the other hand to tag the full EGN-EP genes with the EGN-EP transfer results (/homolog = reyan_gene_id) using the Bedtools (Version 2.26.0) intersect [82,83].

### 4.8. Manual Curation of the Scaffold 1222 Sequence from Clone PB 260

Completeness for scaffold 1222 was checked with clone BPM 24 SMRT reads [19]. Another gene annotation of scaffold 1222 from clone Reyan 7-33-97 was transferred to clone PB 260 by the genome comparison result with RGAAT [84] with default parameters. The analysis of scaffold 1222 was completed by a dot-plot of the Reyan 7-33-97 scaffold against the one for clone PB 260 using Gepard [85]. The schematic representation of the clone PB 260 scaffold 1222 was drawn using DNAPlotter [86].

### 4.9. RNA-Seq Data Analysis

The RSEM program (Version 1.2.29) was used to calculate the normalized expression of full-length transcripts encoding Rubber Elongation Factor (*REF*) and Small Rubber Particle Protein (*SRPP*) found in scaffold 1222 in clone PB 260. As input, we used latex RNA-seq data [39] generated from three healthy trees without ethephon stimulation (water-treated healthy trees, WH), three healthy and three TPD-affected trees subjected to stress induced by ethephon treatment (ethephon-treated healthy trees, EH, and ethephon-treated TPD-affected trees, ET, respectively).

### 4.10. Promoter Analysis

A region of 2 kb was extracted, when possible, from the start codon for *REF1, REF2, REF3, REF4, REF5, REF7, REF8, SRPP1* and *SRPP5.* The PlantCARE webservice (http://bioinformatics.psb.ugent.be/webtools/plantcare/html/) was used to detect *cis*-acting regulatory elements [87].

### 4.11. Phylogenetic Analysis

Full-length protein sequences of the *SRPP/REF* family were extracted from clones PB 260, Reyan 7-33-97 [4] and *Manihot* esculenta (https://phytozome.jgi.doe.gov/pz/portal.html). The MUSCLE program (Version 3.8.31) was used for multiple sequence alignment. The gaps in the alignment were removed using the trimAl program with “–nogaps” (Version 1.4). MrBayes (V3.2.6 x64) was used to build the phylogenetic tree.

### 4.12. Comparative Genome Analysis

Genomic fragments were compared using Gepard [85]. PB 260 scaffold1222 was compared to the clones Reyan 7-33-97 (scaffold lvxx01001222), RRIM600 (scaffold mkxe01001735) and GT1 (chromosome 9 position 13,437,700-13,754,119 bp).

## 5. Conclusions

Altogether, these analyses indicated first of all the importance of accurate TE annotation as a priority in plant genome sequencing projects to minimize the inaccuracy of gene annotation and facilitate functional gene studies. This study confirmed that TEs are the main contributors to genome shaping and, as a consequence, to the regulation of genome expression. These analyses are a valuable resource for comparative epi/genomics within the *Euphorbiacae*, and transposon tagging for designing clone-specific molecular markers in *Hevea.* We had already demonstrated that the natural rubber biosynthesis pathway is under post-transcriptional control [20]. This work shows that this biosynthesis pathway is also under transcriptional control with a major role of TEs, which are functionalized in the promoters of the genes involved in natural rubber biosynthesis.

## Figures and Tables

**Figure 1 ijms-21-04220-f001:**
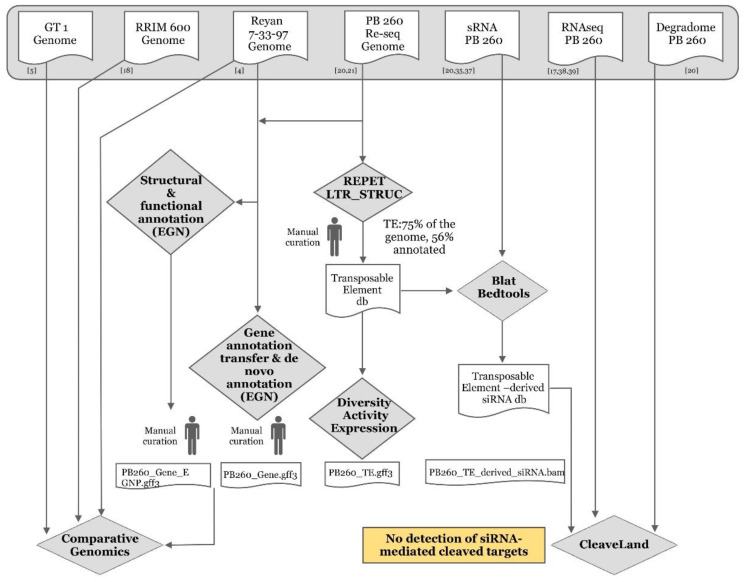
Diagram of the bioinformatics analyses described in this study, and data origins. Numbers refer to references mentioned in the text [4,5,17,18,20,21,35,37,38,39].

**Figure 2 ijms-21-04220-f002:**
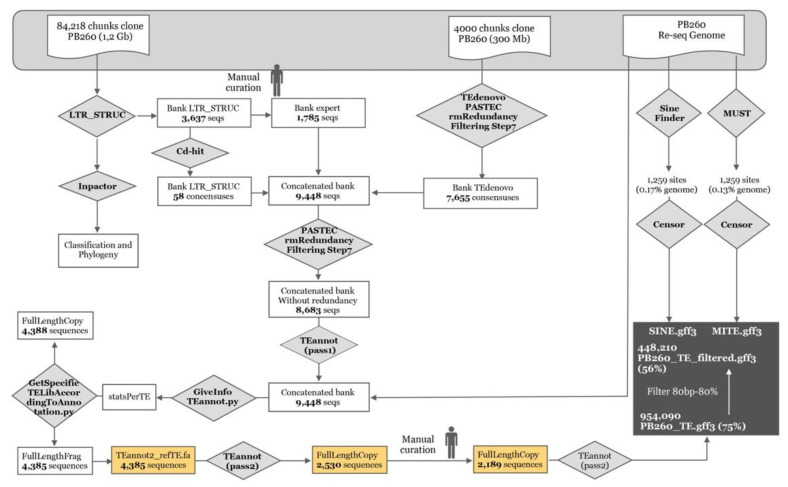
Detailed diagram of the transposable element annotation process implemented by the LTR_STRUCT, REPET pipeline, MUST and Sine_finder.

**Figure 3 ijms-21-04220-f003:**
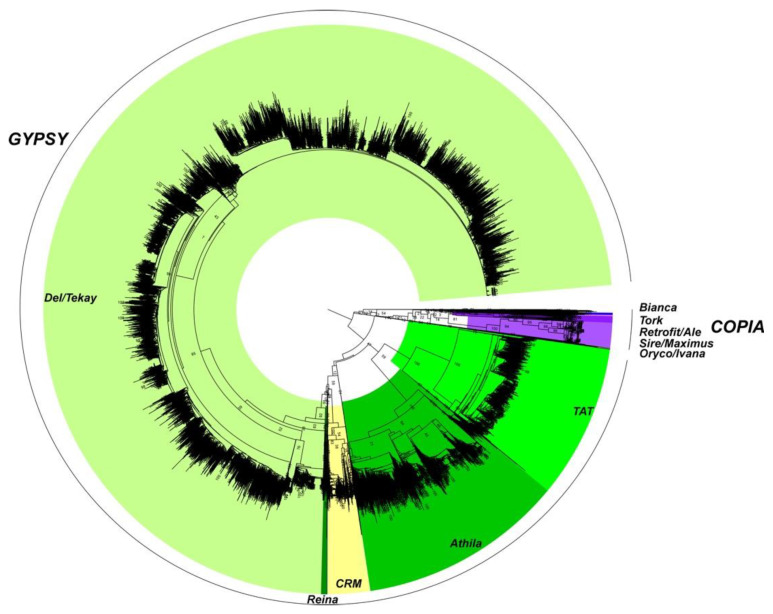
Quantitative representation, on a genome scale, of LTR retrotransposon RT domains in the *H. brasiliensis* PB 260genomic sequences. The phylogenetic analysis was carried out with reverse transcriptase (RT) domains (19,151) having a length > 200 aa and amino-acid identity > 70% compared to the RT reference domains downloaded from GyDB. The *Gypsy* LTR retrotransposon clades are represented in different green and yellow colours. The *Copia* LTR retrotransposon clades are represented in purple.

**Figure 4 ijms-21-04220-f004:**
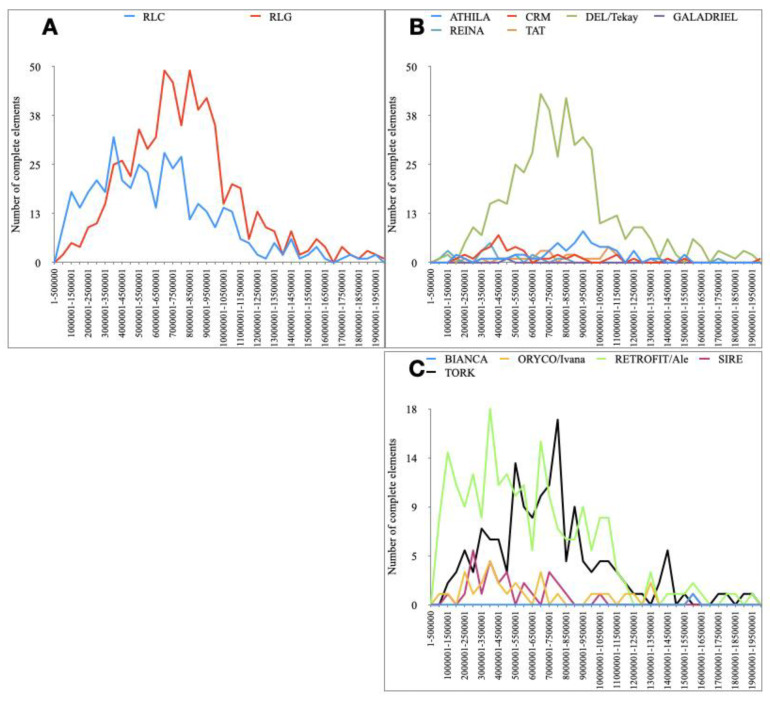
Timing of full-length LTR retrotransposon insertions into the *H. brasiliensis* clone PB 260 genomic sequence. (**A**) Blue and red lines represent, respectively, the number of *Ty1/copia* (RLC) and *Ty3/gypsy* (RLG) full-length LTR retrotransposons per bins of 0.5 Million Years (MY). (**B**) Coloured lines represent the number of full-length *Ty3/gypsy* (RLG) LTR retrotransposon lineages per bins of 0.5 MY. (**C**) Coloured lines represent the number of full-length *Ty1/copia* (RLC) LTR retrotransposon lineages per bins of 0.5 MY. Only the full-length LTR retrotransposons found by LTR_STRUC were used here. An average substitution rate of 3.89 × 10^−9^ was used.

**Figure 5 ijms-21-04220-f005:**
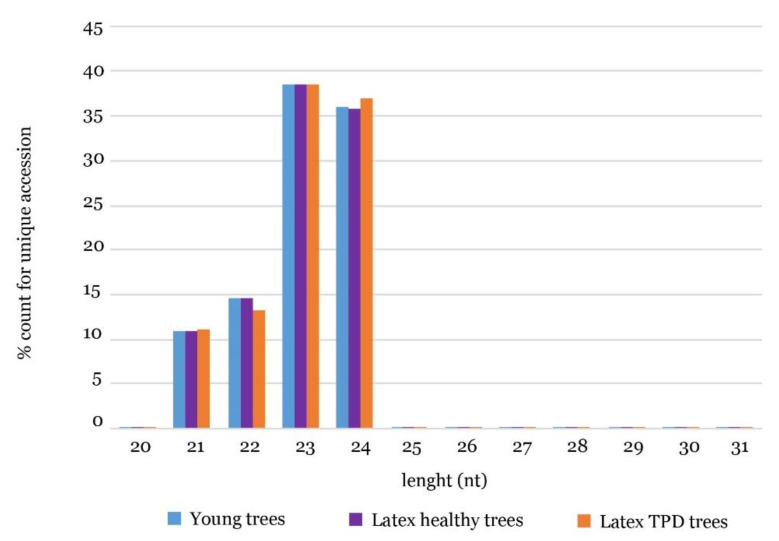
Length distribution of unique TE-derived small RNA accessions from *Hevea* tissues: (young trees (•), latex fromhealthy trees (•) and latex from TPD-affected trees (•)).

**Figure 6 ijms-21-04220-f006:**
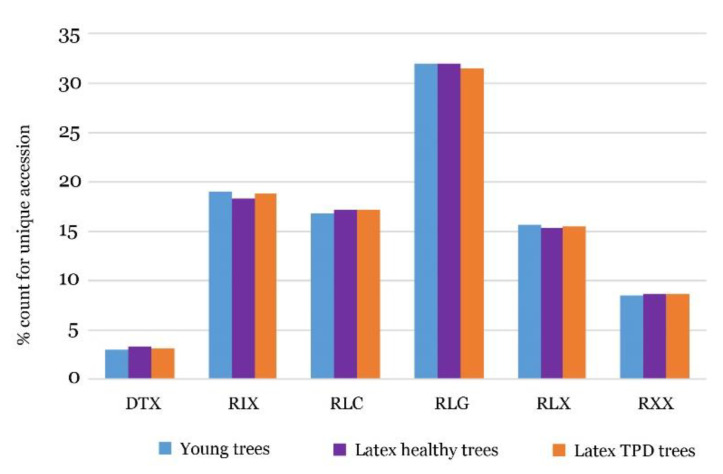
24-nt siRNA production by transposable element order and superfamilies (%).

**Figure 7 ijms-21-04220-f007:**
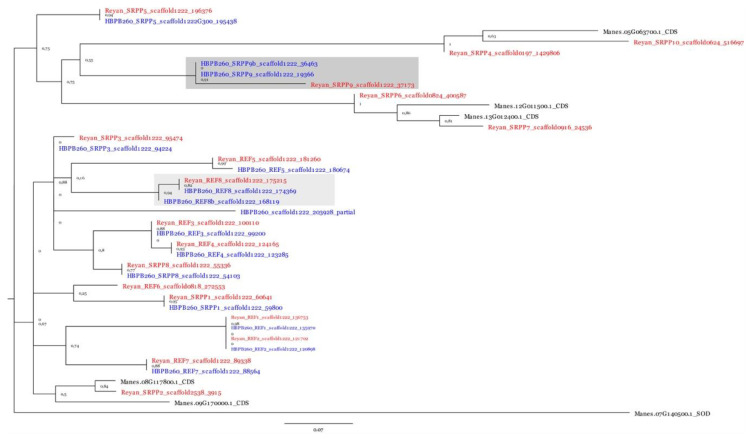
Phylogenetic analysis of the nucleotide sequences of the *REF/SRPP* genes present on scaffold 1222 (Reyan 7-33-97 in red and PB 260 in blue). *Manihot esculenta SRPP* genes were added, as well as a superoxide dismutase gene (*SOD*) as an out-group. Local gene duplication in Reyan 7-33-97 when compared to PB 260 were shaded in light grey.

**Figure 8 ijms-21-04220-f008:**
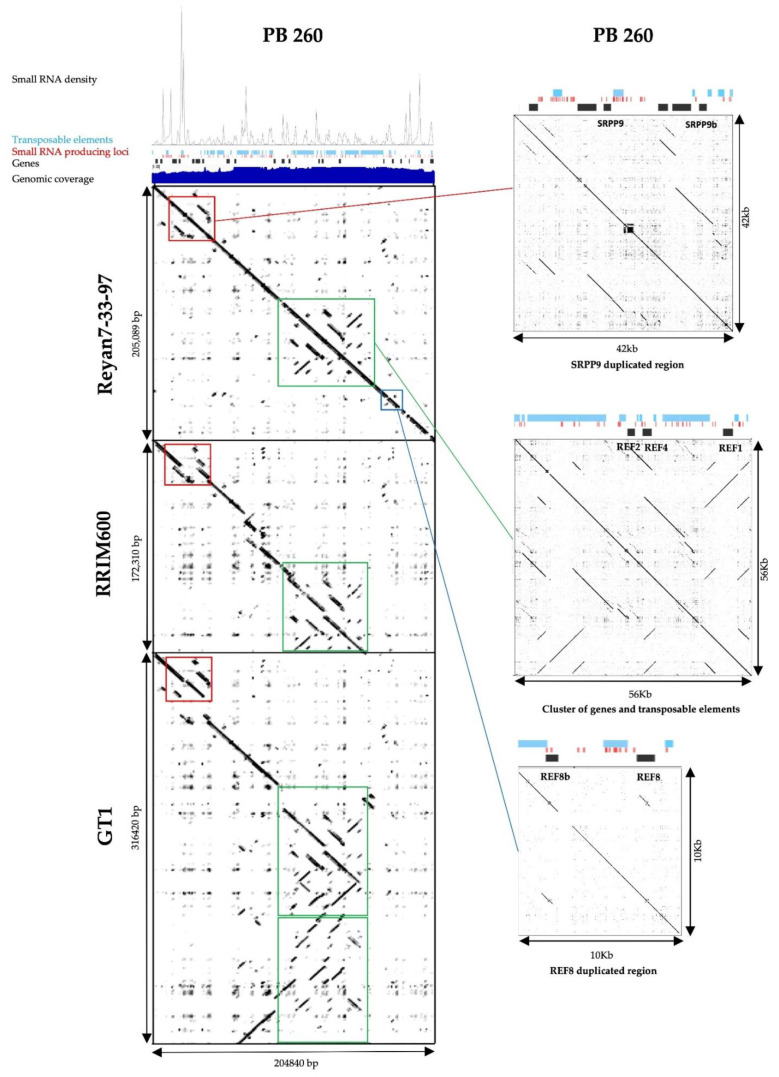
Scaffold 1222 sequence comparison between clones PB 260, Reyan 7-33-97, RRIM 600 and GT 1 by Dot-Plot and genomic DNA read abundance, gene, transposable elements and small RNA density (for PB 260 only). Genomic DNA reads are represented in dark blue, genes in black, transposable elements in light blue, and small RNA in red associated with their density profile. Regions of interest were zoomed in and displayed at the right.

**Table 1 ijms-21-04220-t001:** Statistics of TE annotation of genomic sequences from clone PB 260 by REPET according to Wicker’s classification [25].

PB 260 V1 Assembly	Before Filtering		After Filtering (>79 bp & >79% id)	
TE Lineage	Number	%	Number	%
HX-incomp	1459	0.15	279	0.06
DTX-comp	699	0.07	150	0.03
DTX-incomp	51,214	5.37	20,033	4.47
RIX-comp	2251	0.24	1476	0.33
RIX-incomp	13,529	1.42	10,418	2.32
RLX-comp	86,648	9.08	55,737	12.44
RLX-incomp	587,149	61.54	292,899	65.35
RXX	1902	0.20	1277	0.28
RXX-LARD	122,504	12.84	31,394	7.00
RXX-TRIM	7877	0.83	4162	0.93
Total TE length (bp)	875,232			417,825
Total TE (% genome)	74.49			56.55

**Table 2 ijms-21-04220-t002:** Normalized read counts for REF and SRPP transcripts present on scaffold 1222 from latex of clone PB 260 calculated from RNA-seq data for three independent biological replications on healthy trees [39].

Gene Name	Position (bp)	R1	R2	R3
*SRPP9*	19,366–20,632	51.99	31.99	49.00
*SRPP9b*	36,463–37,765	14.01	13.01	5.00
*SRPP8*	54,103–56,474	108.00	126.02	168.00
*SRPP1*	59,800–61,252	26,2376.06	224,480.10	120,968.96
*REF7*	88,564–89,747	68,916.89	51,496.25	52,838.13
*SRPP3*	94,224–95,735	16.95	17.12	106.20
*REF3*	99,200–100,584	63,213.40	65,499.08	49,443.43
*REF2*	120,898–122,073	356.01	435.57	509.70
*REF4*	123,285–124,611	10,012.49	8702.42	15,524.53
*REF1*	135,970–137,141	509,705.38	452,203.00	392,969.38
*REF8b*	168,119–168,886	1967.65	1328.53	2174.31
*REF8*	174,369–175,626	6423.97	5897.50	4242.77
*REF5*	180,674–181,548	713.99	786.00	568.00
*SRPP5*	195,438–196,845	689.00	978.98	308.00
*partial*	203,928–204,179	0.00	0.00	0.00

**Table 3 ijms-21-04220-t003:** Promoter sequence analysis of PB 260 scaffold1222 with the PlantCare tool (http://bioinformatics.psb.ugent.be/webtools/plantcare/html/). All the *cis*-regulatory elements found are listed and recorded.

Response to	Motif	Element’s Name	*REF1*	*REF2*	*REF3*	*REF4*	*REF5*	*REF7*	*REF8*	*SRPP1*	*SRPP5*
Enhancer	CAAT-box	common cis-acting element in promoter and enhancer regions	4	3	2	2	3	2	4	14	2
light	G-Box	cis-acting regulatory element involved in light responsiveness	1	4	2	2	2	0	0	3	0
	ACE	cis-acting element involved in light responsiveness	1	0	0	0	1	1	1	0	0
	Box 4	part of a conserved DNA module involved in light responsiveness	1	1	8	9	2	0	2	0	1
	MRE	MYB binding site involved in light responsiveness	0	0	1	0	0	0	1	0	0
	GT1-motif	light responsive element	1	0	2	0	0	0	3	0	2
	TCT-motif	part of a light responsive element	0	0	1	1	1	0	0	0	0
	TCA-element	cis-acting regulatory element involved in light responsiveness	0	0	1	1	0	0	0	0	0
	AT1-motif	part of a light responsive module	0	0	0	0	1	0	0	0	0
Anaerobic	ARE	cis-acting regulatory element essential for the anaerobic induction	2	1	2	2	3	1	3	4	1
Hormone	ABRE	cis-acting element involved in abscisic acid responsiveness	0	1	3	1	1	0	0	1	0
	AuxRR-core	cis-acting regulatory element involved in auxin responsiveness	0	0	1	1	0	0	0	0	0
	TGA-element	auxin-responsive element	0	0	1	1	1	1	1	0	1
	AE-box	cis-acting element involved in salicylic acid responsiveness	0	0	1	1	0	0	0	0	0
	CGTCA-motif	cis-acting regulatory element involved in MeJA responsiveness	0	0	0	0	0	0	1	0	0
Abiotic stress	LTR	cis-acting element involved in low-temperature responsiveness	0	0	0	0	0	0	0	2	0
	TC-rich repeats	cis-acting element involved in defence and stress responsiveness	0	0	0	0	0	0	0	1	0
	MBS	MYB binding site involved in drought inducibility	0	0	0	0	0	0	0	1	0
Development	GCN4_motif	cis-regulatory element involved in endosperm expression	0	0	0	0	0	0	0	1	0
	circadian	cis-acting regulatory element involved in circadian control	0	0	0	0	0	0	0	1	0
Others	A-box	cis-acting regulatory element	0	0	0	0	0	0	0	1	0
	O2-site	cis-acting regulatory element involved in zein metabolism regulation	0	0	0	0	0	0	1	0	0
		**Total**	**10**	**10**	**25**	**21**	**15**	**5**	**16**	**29**	**7**
		**Promoter size analysed (kbp)**	**1.791**	**1.222**	**2.011**	**2.011**	**0.736**	**0.493**	**2.001**	**2.001**	**1.715**

**Table 4 ijms-21-04220-t004:** Overview of genome assembly available and used in this study.

	Clones	PB 260 ^1^	Reyan-7-33-97 ^1^	RRIM 600 ^1^	GT 1 ^1^	BPM 24 ^1^
**Available data**	**Genome Assembly**	PRJCA001333	LVXX01000000	AJJZ00000000	PRJNA587314	BDHL00000000
	**WGS data**	PRJCA001333	no	AJJZ00000000	PRJNA587314	Link1
	**SMRT reads**	no	no	no	no	Link2
	**RNA-seq**	PRJCA001333	SRP069104	no	PRJNA587314	no
	**smRNA-seq**	PRJCA001333	no	no	no	no
**Experiments described in this study**	**TE annotation**	this study	[4]	[18]	[5]	[19]
	**siRNA quantification**	this study	no	no	no	no
	**Reconstruction of the SRPP/REF locus**	this study	[4]	[18]	[5]	[19]

**^1^** PB 260 data was deposited to NGDC and other data was deposited to NCBI. Link1 refer to http://www4a.biotec.or.th/rubber/GenomeSeq. Link2 refer to http://www4a.biotec.or.th/rubber/GenomeSeq.

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
