# Peer review of "Structural and Functional Annotation of Transposable Elements Revealed a Potential Regulation of Genes Involved in Rubber Biosynthesis by TE-Derived siRNA Interference in *Hevea brasiliensis"

_ijms, 2020, doi:10.3390/ijms21124220_

Round 1

Reviewer 1 Report

The manuscript deals with the fine characterization of transposable elements and their effect on transcriptional regulation in Hevea brasilensis thourgh TE-derinved siRNA.
The analysis is reasonably well carried out and results support Author’s conclusions.
I enjoyed the reading, though I am not particularly in the field of gene expression and micro RNAs. I cannot find major flaws in the manuscript, but I think a few changes are required in order to accept it for publication.

Page 2, Line 68: change “two” with “terminal” (Terminal Inverted Repeats, actually)

Page 3, Lines 99 and 111: pipeline is named differently.

Page Line 163-164: it could be a good idea to actually show it, maybe in Supplementary material.

Figure 5 can be coloured: greyscale tones do not allow good reading and understanding

Page 7, Line 188: I think something is missing in this sentence. Please, check.

Line 239: better to show it.

Author Response

Dear reviewer,

Many thanks for your comments.

We added some information in order to clarify:

- the interest to do a deep study on one clone (PB 260) and to compare the genomic environment with other clones

-all data used in this study and why

-we modified mostly all the figures for clarity

-we corrected some typos and language errors

We hope that all those changes are helpful for the readers.

The manuscript deals with the fine characterization of transposable elements and their effect on transcriptional regulation in Hevea brasilensis through TE-derived siRNA.
The analysis is reasonably well carried out and results support Author’s conclusions.
I enjoyed the reading, though I am not particularly in the field of gene expression and micro RNAs. I cannot find major flaws in the manuscript, but I think a few changes are required in order to accept it for publication.

Page 2, Line 68: change “two” with “terminal” (Terminal Inverted Repeats, actually)

Reply: Many thanks for your comments, you are right, and corrected as requested.

Page 3, Lines 99 and 111: pipeline is named differently.

Reply: Many thanks for your comments, we redo Figure 1 and replace REPET by REPET/LTR_STRUCT pipelines. In fact, they are presented into details in Figure 2 and generated a misunderstanding. In addition, we made also some modifications to introduce our comparative genomic analyses as requested by reviewer 2

Page Line 163-164: it could be a good idea to actually show it, maybe in Supplementary material.

Reply: Many thanks for your comments, we have prepared a supplementary data with the count tables and the Deseq2 analyses. Three TEs were detected and manual checking showed that their sequences did not correspond to TE (HBPB260_TE7449 HBPB260_TE7754, HBPB260_TE8004). We also change the numeration for other Supplementary data accordingly.

Figure 5 can be coloured: greyscale tones do not allow good reading and understanding

Reply: Many thanks for your comments, we modified it accordingly as well as Figure 6. We modified both of them to answer to reviewer 2.

Page 7, Line 188: I think something is missing in this sentence. Please, check.

Reply: Many thanks for your comments, we have added the missing part in the manuscript to make it clear.

“The production of 24-nt siRNA by TE superfamilies was then quantified by counting those that mapped on TE sequences (Figure 6, Supplementary data 3). The LTR retrotransposon order produced the largest number of siRNAs (70%), with 33% of them by Ty3/gypsy -RLG, followed by unclassified RIX (18%) and Ty1/copia RLC (16%). The DTX of DNA transposons contributed to about3% of siRNA production. We further analysed the 24-nt siRNAs production by TEs in three small RNA-seq libraries including young plants, latex from healthy trees and latex from TPD-affected trees  [34, 36]. Interestingly, there was still no significant in siRNAs production in the three data sets. The same proportions were observed for 21, 22 and 23 nt (Supplementary Figure 3).”

Line 239: better to show it.

Reply: Many thanks for your comments, we referred to supplementary data 1.

Reviewer 2 Report

Using genome assemblies, small RNA datasets, and RNA-seq of the rubber tree Hevea brasiliensis, the authors analyze the interplay between the genomic environment and the silencing/expression of genes in the rubber biosynthetic pathway. For this, they annotate all TEs in a de novo fashion and analyze their coverage with small RNAs. This knowledge is then transferred to the SRPP/REF gene cluster, which is presumably involved in the rubber synthesis. Expression differences among clones are then correlated with the TE/siRNA landscape.  

The manuscript’s topic is timely and interesting, making use of the latest sequencing technologies and – to my knowledge – has not been published so far.

However, the manuscript is not structured very clearly, not all concepts needed are introduced, and thus, it is very hard to understand. I cannot assess the discussion as I did not fully understand the analysis (what was done with each clone? Is a comparison of datasets possible?), but could imagine re-assessing after a very major revision. Therefore, I am omitting the discussion, but try to be as helpful as possible with the rest.

Major points:

1. Data from different rubber tree clones goes into this analysis. As the comparison of the clones is at the heart of the manuscript’s argument, these should be properly introduced. At first, only the clones PB 260 and Reyan 7-33-97 are mentioned, but very late in the results, two additional lines are analyzed. One of the clones, BPM 24, is mentioned in the discussion for the first time. It has to be clarified, why these clones have been included at seemingly random points during the narrative. Generally, the different clones, the switching between them for different analyses, and their general properties remained unclear. This should be clarified throughout the manuscript. To give more detail, I suggest the following:

a) In the abstract I suggest to avoid giving scaffold/clone detail such as “genes are clustered on a scaffold1222 of the genomic sequence of Reyan  7-33-97” or “In clone PB 260, 11% of genes are located near TEs”. This is very confusing and may not be needed. Instead, it may be possible to generalize this for rubber; e.g. “In Hevea genomes, REF/SPRR genes are often clustered at the same locus”, etc.

b) In the introduction section, I miss an introduction of rubber tree plantations. Of special note would be that rubber is clonally propagated, and that the clones differ in yield, rubber properties, etc. The four(?) clones analyzed should be shortly introduced, along with available datasets. Why is it important to investigate different clones and to produce additional datasets? Do these clones have different phenotypic properties?

c) In the results (or M&M), I miss an overview of the four (?) clones, e.g. as a table, with headers such as Available data (subheaders: Genome Assembly, WGS data, SMRT reads, RNA-seq, etc.) and Experiments here (subheaders: TE annotation, siRNA quantification, reconstruction of the SRPP/REF locus, etc.), along with the SRA numbers and references needed.

d) In the flow chart describing the bioinformatics, some of the clones analyzed, e.g. in Figs. 9/10 are not included; this increases the confusion.

e) Why are many of the experimental/bioinformatics analyses only conducted for one of the four clones (but not always the same)? How can the resulting data be interpreted?

f) In the discussion, I miss a straight, easy-to-understand summary before going into detail, e.g. “We re-constructed the SRPP/REF locus in four rubber clones, and found…”. Currently, I cannot evaluate the discussion, as I simply cannot follow the analysis and the argumentation.

2. I have problems confirming the key finding of the manuscript: “Structural and functional annotation of transposable elements revealed a regulation of genes involved in rubber biosynthesis by TE-derived siRNA interference in Hevea brasiliensis”. For this, it may be possible, that this was lost on me due to the difficult organization of the manuscript. This adds to point 1, that stringent structuring of the manuscript is necessary.

Other points:

3. Page 2, lines 65 cont’d.: Why are TE taxa introduced that do not occur in plants, e.g. DIRS?

4. There are some minor problems with the TE terminology:

a) The TE abbreviation “TIR” (terminal inverted repeat) is wrong.

b) In some instances, the TE order “LTR retrotransposons” is referred to as “superfamily”.

5. Page 2, line 92, “our objective  was  to  annotate [..]  TEs and TE-derived  siRNAs precisely”: As far as I can tell, a precise “siRNA annotation” does not exist. In contrast to miRNAs, which are derived from miR genes and can thus be precisely annotated, siRNAs are diced from several transcripts. Thus, exact positions cannot be deduced.  

6. The calculation of LTR retrotransposon integration times is usually restricted to LTR retrotransposons flanked by target site duplications (TSDs). This ensures that the integration is indeed real and not a recombination product, which would lead to loss of the TSDs. Here, in this report, I did not find any statement regarding the consideration of TSDs. If the authors want to consider elements with and without TSDs, it should be made clear, how the results differ. The observed tendencies for integration periods may be biases introduced by recombinational mechanisms, other types of rearrangements, or even the result of an incorrect assembly. The authors should clarify and explicitly state the limitations of their analysis. Alternatively, I would suggest to check for TSDs and to perform the analysis as is standard in the field.

7. Fig. 4 lacks axis labels.

8. The siRNA analysis can only generate meaningful results (comparison 21/24 nt siRNAs), if similar analysis are conducted for both siRNA lengths.

a) The 24 nt siRNAs were mapped to the TEs to determine where they came from. This should also be done for the 21 nt siRNAs (PTGS).

b) The 21 nt siRNAs were localized in the “degradome”. This should also be done with the 24 nt siRNAs.

9. Page 8, line 218 cont’d, Fig.7, “a phylogenetic analysis was carried out with the full-length nucleotide sequences of REF/SRPPs. It revealed local duplication of the SRPP9 and REF8 genes and a partial gene at the distal part of scaffold 1222 (Figure 7).”: It would benefit the manuscript, if the events described were marked in the figure.

10. Table 2 is hard to read and interpret. It would help, if the gene name was dissociated from the position of the scaffold and if the gene entries were sorted alphabetically.

11. Table 3 refers to which clone?

12. It is commendable that the authors represent synteny of the SRPP/REF locus of the four clones by a dotplot. However, currently, one clone is compared to the other at first (fig 8), and then additional clones (which appear out of nowhere) are compared (Fig 9, 10, 11). All to all comparisons are missing. The chosen graphs seem random. The consequences of the structural variation is unexplained. Instead, it would be more consistent, if all four regions are integrated in a multi-dotplot (4x4 matrix; as far as I know, the software Gepard, which the authors used, supports this. Alternatives may be EMBOSS’ polydot, dotter, or Flexidot.)

13. The dotplot figures are labeled problematically:

a) They partially lack axis labels (bp).

b) Fig. 10 lacks the scale completely. Are these zoomed regions?

c) The arrows in Fig. 10 are unclear.

d) The circles in Fig. 9 are unclear.

e) Currently, it is unclear to the reader what to look for in the dotplots. A clear demarcation of the genes, e.g. by shaded regions (as for example generated by Flexidot dotplots) would be helpful.

14. The results sections ends abruptly, without any indication what the results may indicate: “In addition to differential insertion of TEs, a large duplication encompassing different SRPP and REF genes was present (Figure 10 A and B).” It would benefit the manuscript to place the results in a context before entering the discussion.

Minor remarks:

15. Abbreviations in the abstract are partially not explained. This should be remedied.

16. Page 1, line 24: It is unclear, what “their” refers to.

17. Page 2, line 47, “In addition, clonal discrepancies exit.”: Unclear sentence.

18. The manuscript contains many typos, restricted to some sections. But in these sections it is so much that it affects the overall quality of the manuscript, e.g. three typos in the first sentence.

Round 2

Reviewer 2 Report

I want to thank the authors for a thoroughly revised manuscript and also for explaining their changes in a well-addressed letter. The authors have clarified many of my issues. Now, as the results are much clearer, I can also assess the discussion section, and the manuscript as a whole:

Major points

1. In my opinion, the claims of the title and the abstract are overstated. The presented evidence for gene regulation by siRNA is circumstantial. The manuscript cannot clearly demonstrate that the REF/SRPP genes were regulated by TE-derived siRNA interference. For this, a functional analysis would have been needed. Nevertheless, the results are valuable. In my view, toning down the title and abstract section would provide a less misleading framing of the article. The manuscript already states this in the discussion section, and presents an outlook: “In rubber clone PB 260, the next step will be to functionally validate the link between chromatin marks, the presence of TEs, and siRNA production in the environment of genes to demonstrate the transcriptional control of gene expression.” – as this functional evidence is not yet there, the manuscript title has to be changed. Similarly, the following phrase from the abstract is not supported by the results: “We showed that rubber biosynthetic genes were under strong transcriptional regulation by TEs and TE-derived siRNAs.”

Other points

2. The TEs were annotated globally in the genome. For an in-depth analysis, the authors focused on scaffold 1222 carrying the REF and SRPP genes, for which they collected transcriptome data, identified the gene positions and potential promoters, as well as checked structural rearrangements between clones. As the manuscript focuses on the role of TEs and TE-derived siRNAs for this region, I am missing an analysis/ a figure of TEs and siRNA on scaffold 1222 (e.g. a genome browser view, or similar). Otherwise, the global TE/siRNA results cannot be brought into the context of the other, scaffold-specific analyses.

Minor points

3. The newly inserted figures have a very low resolution and can barely be read. This should be remedied.

4. TIR is short for “Terminal inverted repeat”, not “terminal inverted tandem repeat”

5. Page 8, line 224: “Interestingly, there  was  still  no  significant  in  siRNAs  production  in  the  three  data  ” – The sentence is unclear.

6. Page 15, line 29: “We finely annotate transposable elements and identify siRNAs sharing sequence identity with TEs, which a deeper analysis on scaffold 1222 containing a cluster of genes involved in rubber biosynthesis.” I do not understand this sentence. Please improve the grammar.

7. The title of Figure 5 still has the “lenght” typo.

Author Response

I want to thank the authors for a thoroughly revised manuscript and also for explaining their changes in a well-addressed letter. The authors have clarified many of my issues. Now, as the results are much clearer, I can also assess the discussion section, and the manuscript as a whole:

Major points

  1. In my opinion, the claims of the title and the abstract are overstated. The presented evidence for gene regulation by siRNA is circumstantial. The manuscript cannot clearly demonstrate that the REF/SRPP genes were regulated by TE-derived siRNA interference. For this, a functional analysis would have been needed. Nevertheless, the results are valuable. In my view, toning down the title and abstract section would provide a less misleading framing of the article. The manuscript already states this in the discussion section, and presents an outlook: “In rubber clone PB 260, the next step will be to functionally validate the link between chromatin marks, the presence of TEs, and siRNA production in the environment of genes to demonstrate the transcriptional control of gene expression.” – as this functional evidence is not yet there, the manuscript title has to be changed. Similarly, the following phrase from the abstract is not supported by the results: “We showed that rubber biosynthetic genes were under strong transcriptional regulation by TEs and TE-derived siRNAs.”

reply: Thank you for these comments. We toned down the title as follow “Structural and functional annotation of transposable elements revealed a potential regulation of genes involved in rubber biosynthesis by TE-derived siRNA interference in Hevea brasiliensis”. We also modified the sentence from the abstract for clarity as follow: “We hypothesized that the genomic environment of rubber biosynthesis genes has been shaped by TE and TE-derived siRNAs with possible transcriptional interference on their gene expression.”

Other points

  1. The TEs were annotated globally in the genome. For an in-depth analysis, the authors focused on scaffold 1222 carrying the REF and SRPP genes, for which they collected transcriptome data, identified the gene positions and potential promoters, as well as checked structural rearrangements between clones. As the manuscript focuses on the role of TEs and TE-derived siRNAs for this region, I am missing an analysis/ a figure of TEs and siRNA on scaffold 1222 (e.g. a genome browser view, or similar). Otherwise, the global TE/siRNA results cannot be brought into the context of the other, scaffold-specific analyses.

reply: Thank you for this comment. At the top of figure 8, for clone PB 260, we put all information available as small RNA density, transposable elements, small RNA producing loci, genes and genomic coverage. As we merged Figure 8, 9 and 10 together, may be those information are less visible. We will improve the resolution of this figure in order to help the reader.

Minor points

  1. The newly inserted figures have a very low resolution and can barely be read. This should be remedied.

reply: Thank you for this comment. All new inserted figures have been converted to image with a high resolution.

  1. TIR is short for “Terminal inverted repeat”, not “terminal inverted tandem repeat”

reply: Thank you for this comment. This mistake was corrected accordingly.

  1. Page 8, line 224: “Interestingly, there  was  still  no  significant  in  siRNAs  production  in  the  three  data  ” – The sentence is unclear.

reply: Thank you for this comment. In a previous study, we showed a change in small RNA distribution between young plantlet and latex of healthy trees compared to TPD-affected trees (Gébelin et al 2013). Recently, we showed that this change of the small RNA distribution was not explainable by a change in miRNA distribution (Leclercq et al 2020). Here, we demonstrated that the change in small RNA distribution is not due to siRNA sharing sequence identity with TE. So, to be clearer, we changed the sentence:

“Interestingly, there was no significant in siRNAs production by transposable elements between young plantlets, latex from healthy and TPD-affected trees”.

  1. Page 15, line 29: “We finely annotate transposable elements and identify siRNAs sharing sequence identity with TEs, which a deeper analysis on scaffold 1222 containing a cluster of genes involved in rubber biosynthesis.” I do not understand this sentence. Please improve the grammar.

reply: Thank you for this comment. We corrected the grammar as follow: “We precisely annotated the transposable elements to identify all siRNAs sharing a sequence identity with them. Further analysis was performed on scaffold 1222 containing a cluster of genes involved in rubber biosynthesis”.

  1. The title of Figure 5 still has the “lenght” typo.

reply: Thank you. This mistake has been corrected.